# Is Alexithymia a Trait or a State? Temporal Stability in a Three-Wave Longitudinal Study

**DOI:** 10.3390/jcm14082628

**Published:** 2025-04-11

**Authors:** Paweł Larionow, Karolina Mudło-Głagolska, David A. Preece

**Affiliations:** 1Faculty of Psychology, Kazimierz Wielki University, 85-064 Bydgoszcz, Poland; mudlo@ukw.edu.pl; 2Institute of Psychology, University College of Professional Education in Wroclaw, 53-329 Wrocław, Poland; 3School of Population Health, Curtin University, Perth, WA 6102, Australia; david.preece@curtin.edu.au; 4Department of Psychology, Stanford University, Stanford, CA 94305, USA; 5School of Psychological Science, The University of Western Australia, Perth, WA 6009, Australia

**Keywords:** absolute stability, alexithymia, anxiety, depression, psychopathology, relative stability, somatic symptoms, temporal stability, test–retest, well-being

## Abstract

**Background/Objectives**: Alexithymia is a construct involving deficits in the cognitive processing of emotions. It is often regarded as an important risk factor for a variety of psychopathologies, but there is ongoing uncertainty about whether it might act as a stable trait, or if instead it is just a state reaction to distress. Our aim was to examine absolute and relative stability of alexithymia over a period of 7 months. **Methods**: Our sample included 73 general community adults who, over three time points, completed the Perth Alexithymia Questionnaire-Short Form (PAQ-S) and a battery of self-report questionnaires assessing several key mental health outcomes (i.e., anxiety and depression symptoms, somatic complaints, well-being). **Results**: Our results showed strong absolute stability for PAQ-S scores, with no significant differences in alexithymia levels between the three time points (*p* > 0.05, η^2^ = 0.01, ω^2^ = 0.00), whereas there were differences in the scores of other mental health outcomes. We also demonstrated good relative stability of alexithymia, with its scores being independent of changes in the other health outcomes. **Conclusions**: Our findings therefore support the capacity for alexithymia to act as a stable trait. As such, our results help to further uncover the nature of alexithymia and support its status as a risk factor for poor affective outcomes.

## 1. Introduction

Alexithymia is a construct involving deficits in the cognitive processing of emotions, including difficulties identifying, describing and focusing attention on one’s own feelings [1]. Since the first observations of alexithymia among psychiatric patients in the 1970s [2], it has often been conceptualized as a relatively stable trait that operates as a transdiagnostic risk factor for a wide range of mental health problems. Indeed, alexithymia levels have been found to predict various psychopathologies, including depression, anxiety, eating disorders, substance use, personality disorders and psychosomatic disorders (e.g., Picardi et al. [3], Westwood et al. [4], Hemming et al. [5]). Conceptual models suggest that this may be because alexithymia impairs emotion regulation, predisposing people to dysregulated levels of affect [6]. There is also evidence that alexithymia can impair treatment effectiveness in psychotherapy settings [7]. Thus, the alexithymia construct is of high clinical interest. However, one area of ongoing uncertainty is whether alexithymia is indeed a stable trait that may predispose people to mental illness, or if instead alexithymia might just be a state-based phenomenon involving reactions to distress. That is, the extent to which alexithymia levels might fluctuate over time is unclear.

Contemporary affective science frameworks like the *attention appraisal model of alexithymia* [8] suggest that alexithymia may have both trait and state elements. From this perspective, alexithymia should be influenced by the *developmental level of people’s emotion schema systems* (e.g., one’s theoretical knowledge of affect), which will dictate the levels of ability to process emotions [8]. One would expect these emotion schema systems to be relatively stable, outside of being directly targeted with interventions. At the same time though, within this framework, alexithymia levels are also likely to be influenced by the *extent to which one is avoiding focusing on their emotions*. This aspect might be expected to fluctuate more readily, as if someone is going through a period of distress, they may try to disengage from their emotions as a coping response, manifesting as higher alexithymia [9].

Over the past few decades, a range of studies have examined the stability of alexithymia. There are two types of temporal stability of relevance here: *absolute* and *relative* stability [10]. Absolute stability refers to the extent to which alexithymia scores change over time (i.e., a comparison of participants’ alexithymia scores across different time points, usually assessed by paired *t*-tests or analyses of variance [ANOVAs]). In contrast, relative stability refers to the extent to which the relative differences among participants in a sample remain the same over time (i.e., test–retest stability, usually examined with Pearson correlation coefficients and intraclass correlation coefficients [10]). In this sense, absolute stability is a stricter criterion than relative stability.

As noted in a systematic review by Karukivi and Saarijärvi [11], generally, alexithymia studies on adults have found high relative stability though mixed results regarding absolute stability. For example, Luminet et al. [10] reported a lack of absolute stability but evidenced good relative stability of alexithymia in patients with major depression during treatment with antidepressant medication. The same pattern was observed in a sample of females with breast cancer, with changes in alexithymia scores being associated with changes in depression scores only to a small extent [12]. In contrast, some other studies have found evidence that alexithymia can be a state-dependent characteristic in patients with major depressive disorder, with changes in alexithymia levels being related to corresponding changes in depression levels [13,14,15].

However, there are several important limitations of the current literature that make this area underexplored. One limitation is that the existing studies of temporal stability have all used the Toronto Alexithymia Scale-20 (TAS-20 [16]) to operationalize alexithymia. The TAS-20 has been a popular measure; however, many recent studies have highlighted several significant psychometric limitations of this tool. For example, the externally oriented thinking items have low reliability [17,18], and of most relevance, the TAS-20 appears to be confounded by people’s current distress levels; that is, the TAS-20 has been found to, in part, assess how distressed people are currently, rather than just their alexithymia levels (e.g., [19,20,21,22]). These features might impair the capacity to accurately capture the temporal dynamics of alexithymia; therefore, there is a need for work with more recent and psychometrically robust measures. Another consideration is that the majority of studies have examined temporal stability in clinical samples, often in the context of undergoing treatment. While such data are useful, given that alexithymia is dimensional and normally distributed across the general population [11,23], there is also an important need to evaluate its stability in non-clinical samples under more typical conditions. Lastly, many of the existing studies have examined alexithymia only at two time points, over relatively short periods (e.g., [3,10,24]), and thus, there is a need for studies that include longer examinations of stability.

Our aim in this study was to further the understanding of the nature of alexithymia, examining its temporal stability (i.e., absolute and relative stability) in a non-clinical sample across three time points over 7 months. We used the Perth Alexithymia Questionnaire-Short Form (PAQ-S [25]) to operationalize alexithymia—a more recent tool, which, along with its long form, has consistently demonstrated strong psychometric performance (e.g., [26,27,28]). Additionally, while most previous studies have only looked at negative mental health indicators (e.g., anxiety and depression symptoms) as potential confounding variables, we broadened this to also include negative physical health indicators (i.e., somatic complaints) and positive mental health indicators (i.e., well-being).

## 2. Materials and Methods

### 2.1. Procedure

The study was conducted according to the ethical principles of the Declaration of Helsinki. Ethical approval for this research was granted by the Kazimierz Wielki University Ethics Committee (No. 1/13 June 2022, later revisions 27 June 2023 and 11 November 2024). The study was anonymous and voluntary, and all participants provided written informed consent for use of their data.

Participants were university students recruited at the university and informed about the survey during classes. Each participant completed a set of psychometric self-report questionnaires (paper and pencil method) across the three waves of the study, memorizing a unique code to link their data. Of the three waves, the first measurement (T1) was conducted in the middle of November 2023, the second (T2) at the end of January 2024 and the third (T3) at the end of June 2024. The break between T1 and T2 was 77 days (11 weeks), and between T2 and T3, it was 141 days (20 weeks). Thus, the total span was around 7 months. At T1, 142 participants filled out the questionnaires; however, not all of these participants were able to complete the survey three times for various reasons (e.g., absence from the university at subsequent time points). Therefore, the attrition rate was about 49%, and our final sample included 73 participants.

### 2.2. Participants

Our participants were 73 Polish-speaking social science students (59 females, 9 males, 2 non-binary and 3 missing data on gender) with ages ranging from 18 to 29 (*M* = 18.96, *SD* = 1.40) from the Kazimierz Wielki University in Poland.

### 2.3. Measures

#### 2.3.1. The Perth Alexithymia Questionnaire-Short Form (PAQ-S)

The PAQ-S [25] is a six-item self-report measure of alexithymia (e.g., “When I’m feeling *bad*, I can’t tell whether I’m sad, angry, or scared”). Items are scored on a seven-point Likert scale ranging from 1 (strongly disagree) to 7 (strongly agree), with higher scores indicating a higher level of alexithymia. All items are summed into a total score. In this study, the Polish version of the questionnaire was used [27].

#### 2.3.2. The Patient Health Questionnaire-4 (PHQ-4)

The PHQ-4 is a four-item self-report questionnaire for assessing anxiety and depression symptoms over the previous two weeks [29,30]. The PHQ-4 has two two-item subscales: anxiety (e.g., “Not being able to stop or control worrying”) and depression (e.g., “Little interest or pleasure in doing things”). Items are scored on a four-point Likert scale, ranging from 0 (not at all) to 3 (nearly every day). Higher scores indicate higher levels of symptoms. In this study, the Polish version of the questionnaire was used [31].

#### 2.3.3. The Giessen Subjective Complaints List-8 (GBB-8)

The GBB-8 [32,33] is an eight-item self-report questionnaire for measuring somatic symptoms (e.g., “Being easily exhausted”; “Feeling bloated or distended”; “Neck or shoulder pain”). A total score, representing general somatic symptom burden, can be calculated. The GBB-8 uses a five-point Likert scale from 0 (not at all) to 4 (very much). Higher scores indicate a higher level of symptoms. In this study, the Polish version of the scale was used [34].

#### 2.3.4. The WHO-Five Well-Being Index (WHO-5)

The WHO-5 [35,36] is a five-item self-report questionnaire for assessing positive well-being. Items (e.g., “I have felt active and vigorous”) are scored on a six-point Likert scale, ranging from 0 (at no time) to 5 (all the time). Higher scores indicate a higher level of well-being. In this study, the Polish version of the WHO-5 was used. The Polish WHO-5 has demonstrated strong psychometric properties in the general population [37].

### 2.4. Statistical Analysis

#### 2.4.1. Preliminary Analysis

Descriptive statistics and frequentist and Bayesian internal consistency reliability coefficients (i.e., McDonald’s omega and Cronbach’s alpha) for the study variables were calculated.

#### 2.4.2. Absolute Stability

To evaluate the *absolute* stability of alexithymia and other constructs between three time points (T1–T3), repeated measures ANOVA with the supplementation of effect size measures (i.e., η^2^ and ω^2^) was conducted. In case of statistically significant differences in scores between the three time points, post hoc comparisons with Bonferroni corrections were computed.

#### 2.4.3. Relative Stability

To evaluate the *relative* stability of alexithymia and other constructs between the three time points, we used four complimentary methods:Pearson correlations between the variables at T1–T3 were computed.Intraclass correlation coefficients between the three time points for the study variables were calculated. Values of less than 0.50 for these coefficients indicate poor stability (i.e., test–retest reliability); values between 0.50 and 0.75 indicate moderate stability; values between 0.75 and 0.90 indicate good stability; and values greater than 0.90 indicate excellent stability [38].Multiple regression analyses were applied to examine the degree to which the alexithymia scores were predicted by the scores of other variables (i.e., anxiety, depression, somatic symptoms and well-being levels).Multiple regression analyses were conducted to examine the degree to which changes in alexithymia scores were related to changes in scores of other variables (i.e., anxiety, depression, somatic symptoms and well-being levels).

In terms of the regressions, the third method was implemented with a hierarchical multiple regression analysis model with 5 steps. In this model, T3 PAQ-S scores served as the dependent variable. In Step 1, GBB-8 scores at the three time points were input. In Step 2, they were supplemented by PHQ-4 Depression scores at the three time points. In Step 3, PHQ-4 Anxiety scores at the three time points were added. In Step 4, WHO-5 scores at the three time points were included. Finally, in Step 5, T1 PAQ-S and T2 PAQ-S scores were input.

The fourth method was implemented with multiple regression analysis, where the difference between T1 and T3 for PAQ-S scores (i.e., T1 PAQ-S scores minus T3 PAQ-S scores) was the dependent variable. The four predictor variables were four difference scores for the difference between T1 and T3 for anxiety, depression, somatic symptoms and well-being scores (i.e., the change in each of these variables over time).

## 3. Results

### 3.1. Descriptive Statistics and Reliability

Descriptive statistics and internal consistency reliability coefficients are displayed in Table 1. Internal consistency reliability was acceptable for all the questionnaire scores across the three time points.

### 3.2. Absolute Stability of Alexithymia and Other Health Constructs

Repeated measures ANOVAs revealed that there were no statistically significant differences in PAQ-S scores between the three time points. There were also no differences in PHQ-4 Depression and GBB-8 scores between the three time points. In contrast, there were statistically significant differences in PHQ-4 Anxiety scores (with anxiety levels at T1 being lower than anxiety levels at T3) and WHO-5 scores (with well-being levels at T1 and T2 being higher than well-being levels at T3). As such, these results indicated that some mental health indicators (i.e., anxiety symptoms and well-being) became less favorable with time, whereas no changes were evident in the levels of alexithymia, depression symptoms and somatic complaints (see Table 1).

### 3.3. Testing the Relative Stability of Alexithymia and Other Health Constructs Using Correlational Analysis

Pearson correlations between the study variables are presented in Appendix A. PAQ-S T1 was strongly positively associated with PAQ-S T2 (*r* = 0.67, *p* < 001) and PAQ-S T3 (*r* = 0.58, *p* < 0.001). PAQ-S T2 and PAQ-S T3 were also strongly positively associated (*r* = 0.56, *p* < 0.001), thus supporting the relative stability of alexithymia. Intraclass correlation coefficients between the three time measurements indicated moderate stability (test–retest reliability) for the PAQ-S scores, as well as moderate stability for the GBB-8 somatic symptom scores and WHO-5 well-being scores and poor stability for the PHQ-4 Anxiety and PHQ-4 Depression scores (see Table 2).

### 3.4. Testing the Relative Stability of Alexithymia Using Multiple Regression Analysis

#### 3.4.1. Examining the Degree to Which the Relative Stability in Alexithymia Scores Is Related to Severity of Other Health Constructs

Table 3 presents the final model (i.e., including Step 5) of the hierarchical multiple regression analysis predicting PAQ-S T3 scores. Among all the included variables, only PAQ-S T2 scores were a significant predictor of PAQ-S T3 scores. Steps 1–4, with the three time points of GBB-8 scores, PHQ-4 Depression scores, PHQ-4 Anxiety scores and WHO-5 scores, were statistically insignificant (see Appendix A), with adjusted R^2^ ranging from 1% to 6% (see Appendix A). Only the addition of PAQ-S T1 scores and PAQ-S T2 scores to the final step improved the model fit, *F*(14,58) = 3.94, *p* < 0.001, with an adjusted R^2^ of 36%. Overall, these results indicated that alexithymia scores at T3 were not predicted by variables outside of prior alexithymia levels, indicating that other variables did not significantly impact T3 alexithymia levels. Thus, the results support the strong relative stability of alexithymia.

#### 3.4.2. Examining the Degree to Which Changes in Alexithymia Scores Are Related to Changes in Other Health Constructs

The multiple regression model predicting changes in the PAQ-S scores, with the four predictors being change scores in the other health variables (i.e., T1–T3 difference scores for PHQ-4 Anxiety, PHQ-4 Depression, GBB-8 and WHO-5), was not statistically significant, *F*(4,68) = 0.66, *p* = 0.625 (see Appendix A). These results suggest that changes in alexithymia cannot be attributed to changes in the other negative and positive health variables.

## 4. Discussion

Our aim here was to explore the nature of alexithymia in terms of examining its temporal stability and whether it acted as a trait or a state in a non-clinical sample over a 7-month period. As assessed by the PAQ-S, we found that alexithymia demonstrated strong stability overall, thus supporting its status as a trait.

In terms of the various types of stability and metrics we examined, we noted strong absolute stability of the PAQ-S scores, with no statistically significant differences in alexithymia scores between the three time points. This contrasted with some of the mental health markers (e.g., anxiety, well-being) that had significant differences, suggesting that alexithymia was more stable than those other constructs and that alexithymia could maintain a good level of stability even when mental health levels shifted.

Our results also strongly supported the relative stability of alexithymia (i.e., the changes in a ranking of one’s scores). Pearson and intraclass correlation coefficients showed that alexithymia was characterized by good relative stability over the three time points, proving to be more stable than anxiety symptoms, depression symptoms and well-being (but less stable compared to somatic complaints in these data). Our regression analyses further evidenced that alexithymia scores at the end of the seven months were uniquely predicted only by earlier alexithymia scores, not the other health variables—a pattern that was maintained when change scores were examined too.

As such, our results are broadly in line with past findings that have typically found good relative stability for the alexithymia construct (e.g., [3,10,12,39,40]). Our findings build on this past work by highlighting that this stability also seems to exist in non-clinical samples, outside of treatment settings, and across relatively long time periods.

Our finding of good absolute stability is rarer (e.g., [41]) and highlights that alexithymia was more stable in our dataset than has been observed in some other studies (e.g., [10,12,40]). On the one hand, one possibility is that the use of the TAS-20 in past work increased instability (i.e., due to the TAS-20 partly measuring current distress levels [19,20,21,22]), but this greater stability could also reflect the non-clinical nature of our sample and the fact that participants were not systematically undergoing mental health treatment [10,12]. Indeed, greater stability has previously been observed in non-clinical as compared to clinical samples (e.g., De Gucht et al. [42]). For example, our results are in line with a similar study conducted with the TAS-20 in a sample of university students by Martínez-Sánchez et al. [24], where, across two time points, they noted shifts in mental health outcomes but stability in alexithymia levels.

In Poland, the prevalence of mental health issues is high [31,37], and in the current sample, the mean PHQ-4 scores were around 3, suggesting that many respondents experienced elevated levels of anxiety and/or depression disorder symptoms. Thus, despite not being a clinical sample, elevated symptoms were still present in a substantial number of participants. Future work examining temporal stability with psychometrically robust assessment tools like the PAQ-S or PAQ in clinical samples would be useful to help disentangle the role that different measurement tools or the nature of the sample may play in the observed stability levels.

Taken together, our findings therefore reinforce the trait status of alexithymia, meaningfully adding to the theoretical knowledge about the nature of the construct. By extension, these findings support the potential utility of continuing to consider alexithymia as a risk factor for the development and maintenance of psychopathology [6,43], rather than just being a state-dependent phenomenon that occurs secondary to distress [9,14]. This is not to say that alexithymia cannot present as a state (as there is evidence of alexithymia levels fluctuating in response to distress [14]) but rather that, in our dataset, it was capable of acting as a trait.

### 4.1. Limitations of the Research

The current research was an observational longitudinal study. As such, we did not have a comparator or an experimental group, which underwent mental health interventions. We did not control for participants’ clinical diagnoses and their treatments, if any existed. Our participants were from a relatively narrow demographic range; they were young adults, mostly female, and all were university (social science) students based in a single European country. As such, our results are most relevant to the young Polish population.

### 4.2. Future Directions

Future studies are needed to examine the stability of alexithymia across more diverse samples, including cross-cultural work with the possibility to explore the potential moderating effects of demographic factors. Adding an experimental group, controlling for clinical diagnoses and controlling for mental health interventions would be fruitful for a more nuanced understanding of clinical circumstances under which alexithymia is less or more stable. It is likely that the stability of alexithymia and/or the magnitude of its potential changes differ in non-clinical, sub-clinical and clinical samples. The measure of alexithymia we used was the brief six-item version of the PAQ (PAQ-S), which was well suited to this repeated assessment context, but it was designed to provide only a total score as an overall marker of alexithymia (rather than subscale scores). Future work might use the long-form 24-item PAQ and its subscales to more comprehensively explore how different facets of alexithymia (across both negative and positive emotions) act over time. It is possible that some elements of alexithymia might be more stable than others (e.g., [42]).

## 5. Conclusions

Our data suggest that alexithymia, as measured by the PAQ-S, can function as a trait with strong stability over time. In turn, our findings enhance the understanding of the nature of alexithymia and support conceptualizations that position alexithymia as an important risk factor for the development and maintenance of poor affective outcomes.

## Figures and Tables

**Table 1 jcm-14-02628-t001:** Descriptive statistics, frequentist and Bayesian internal consistency reliability coefficients for the study variables and the repeated measures ANOVA results (*n* = 73).

Variables	Frequentist McDonald’s Omega	Frequentist Cronbach’s Alpha	Bayesian McDonald’s Omega	Bayesian Cronbach’s Alpha	Mean	*SD*	Repeated Measures ANOVA
PAQ-S T1	0.70	0.68	0.68	0.68	19.34	7.14	*F*(2.00, 144.00) = 0.79, *p* = 0.454, η^2^ = 0.01, ω^2^ = 0.00
PAQ-S T2	0.78	0.78	0.77	0.77	19.27	7.67	
PAQ-S T3	0.84	0.84	0.83	0.83	18.45	7.78	
PHQ-4 Anxiety T1	0.75	0.75	0.71	0.74	3.25	1.65	*F*(2.00, 144.00) = 6.12, *p* < 0.003, η^2^ = 0.08, ω^2^ = 0.02, post hoc (Bonferroni): T1 < T3
PHQ-4 Anxiety T2	0.84	0.84	0.80	0.84	3.56	1.72	
PHQ-4 Anxiety T3	0.71	0.71	0.66	0.70	3.95	1.61	
PHQ-4 Depression T1	0.84	0.84	0.80	0.84	2.64	1.76	*F*(2.00, 144.00) = 2.57, *p* < 0.080, η^2^ = 0.03, ω^2^ = 0.01
PHQ-4 Depression T2	0.85	0.85	0.81	0.84	2.51	1.86	
PHQ-4 Depression T3	0.80	0.80	0.75	0.80	2.97	1.63	
GBB-8 T1	0.83	0.83	0.82	0.83	14.03	6.76	*F*(2.00, 144.00) = 0.70, *p* < 0.499, η^2^ = 0.01, ω^2^ = 0.00
GBB-8 T2	0.85	0.85	0.84	0.84	14.41	7.06	
GBB-8 T3	0.83	0.82	0.82	0.82	14.82	6.49	
WHO-5 T1	0.87	0.87	0.86	0.87	11.25	4.76	*F*(2.00, 144.00) = 10.74, *p* < 0.001, η^2^ = 0.13, ω^2^ = 0.04, post hoc (Bonferroni): T1 > T3, T2 > T3
WHO-5 T2	0.84	0.84	0.83	0.83	10.27	4.44	
WHO-5 T3	0.85	0.85	0.83	0.84	9.01	4.26	

**Table 2 jcm-14-02628-t002:** Intraclass correlation coefficients between the three time measurements (T1–T3) for the study variables (*n* = 73).

Variables	Intraclass Correlation Coefficients with 95% CI	Interpretation of Test–Retest Reliability
PAQ-S	0.60 (0.48, 0.71)	Moderate
PHQ-4 Anxiety	0.45 (0.31, 0.59)	Poor
PHQ-4 Depression	0.46 (0.33, 0.60)	Poor
GBB-8	0.64 (0.53, 0.74)	Moderate
WHO-5	0.55 (0.40, 0.67)	Moderate

Note. Intraclass correlation coefficients were calculated using two-way random effects, absolute agreement, single rater/measurement. Interpretation of test–retest reliability is based on the guidelines by Koo and Li [38].

**Table 3 jcm-14-02628-t003:** Multiple regression analysis coefficients in the final Step 5, with all predictors (*n* = 73).

Predictors	Unstandardized	Standard Error	Standardized	*t*	*p*	Tolerance
(Intercept)	5.04	7.40	–	0.68	0.498	–
GBB-8 T1	–0.07	0.20	−0.06	−0.33	0.742	0.29
GBB-8 T2	0.01	0.22	0.01	0.04	0.969	0.22
GBB-8 T3	0.19	0.19	0.16	1.00	0.324	0.36
PHQ-4 Depression T1	−0.09	0.69	−0.02	−0.13	0.897	0.37
PHQ-4 Depression T2	0.89	0.87	0.21	1.03	0.306	0.21
PHQ-4 Depression T3	−0.57	0.76	−0.12	−0.75	0.456	0.35
PHQ-4 Anxiety T1	0.31	0.80	0.07	0.39	0.697	0.31
PHQ-4 Anxiety T2	−0.96	0.85	−0.21	−1.13	0.262	0.25
PHQ-4 Anxiety T3	0.42	0.71	0.09	0.59	0.558	0.41
WHO-5 T1	0.08	0.28	0.05	0.29	0.775	0.30
WHO-5 T2	0.24	0.34	0.14	0.71	0.480	0.24
WHO-5 T3	−0.54	0.29	−0.30	−1.86	0.068	0.35
PAQ-S T1	0.30	0.15	0.27	1.97	0.054	0.46
PAQ-S T2	0.39	0.14	0.39	2.73	0.008	0.44

## Data Availability

The raw data supporting the conclusions of this article will be made available by the authors on request.

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
