# Peer review of "Is Alexithymia a Trait or a State? Temporal Stability in a Three-Wave Longitudinal Study"

_jcm, 2025, doi:10.3390/jcm14082628_

Round 1
Reviewer 1 Report
Comments and Suggestions for Authors
Is Alexithymia a Trait or State? Temporal Stability in a Three-2 Wave Longitudinal Study
The aim of the present study is to evaluate absolute and relative stability of alexithymia over a period of 7 months. This is a nice topic, and it is worthy of investigation. Here are my comments regarding the strengths of the paper as well as the sections still to be further developed.
Literature
The background literature has mainly been summarized correctly, and the aim of the study has been stated clearly. However, it should be noted that there is no control group included in this study.
Participants
The participants were mostly young female university students. Is it possible to identify previous studies focusing on this age group? Are there any differences between the respondents (those who participated all three times) and the non-respondents? Do the participants have any other notable psychiatric diagnoses besides depression or anxiety disorders? If so, the diagnostic distribution of the sample should be described.
Methods
The study used the Perth Alexithymia Questionnaire-Short Form (PAQ-S) and a battery of 17 self-report questionnaires assessing several key mental health outcomes (i.e., anxiety and 18 depression symptoms, somatic complaints, well-being).
Results
The prevalence of alexithymia of the study population is very low when compared to studies of general populations. This probably shows that our sample represent mostly healthy young adults. The results showed that the relative and absolutely stability of alexithymia was high.
Discussion
The interpretations are justified by the results, and the relevant findings are discussed in relation to prior research. However, a more critical approach is needed, especially considering the sample size and the age and gender distribution. I would also appreciate a more critical examination of the concepts of absolute and relative stability. To what extent can these findings be generalized, and to which population.
Author Response
We would like to thank the editor and the reviewers for their positive and encouraging feedback on our submission. The constructive comments of reviewers helped us to significantly improve the quality of our submission. We have been through all comments one by one, edited the manuscript in detail, and added new material where required. We hope the editor and reviewers find the revised version of the manuscript clear and suitable for publication in the Journal of Clinical Medicine. All changes made are highlighted in red (in the replies and the revised paper).
Reviewer 1
Is Alexithymia a Trait or State? Temporal Stability in a Three-Wave Longitudinal Study
The aim of the present study is to evaluate absolute and relative stability of alexithymia over a period of 7 months. This is a nice topic, and it is worthy of investigation. Here are my comments regarding the strengths of the paper as well as the sections still to be further developed.
Literature
The background literature has mainly been summarized correctly, and the aim of the study has been stated clearly. However, it should be noted that there is no control group included in this study.
Reply: This longitudinal study was an observational one. That is, there was no external impact on our participants, who were mostly young healthy people, therefore, by the nature of this study, no control group was included. We agree with Reviewer 1 that adding another group (e.g., people with clinical diagnoses) would be great for testing stability of alexithymia, and we have now indicated this in the limitations and future directions section of the discussion.
Participants
The participants were mostly young female university students. Is it possible to identify previous studies focusing on this age group? Are there any differences between the respondents (those who participated all three times) and the non-respondents? Do the participants have any other notable psychiatric diagnoses besides depression or anxiety disorders? If so, the diagnostic distribution of the sample should be described.
Reply: Previous studies regarding temporal stability of alexithymia, with a design similar to our one, were basically done in clinical samples undergoing treatments during the observational period (Luminet et al.), which we have already cited. We have found a study by Francisco Martínez-Sánchez et al., who examined temporal stability of alexithymia and other emotional variables in students. We have now elaborated on these findings, and contrasted them with our ones in the discussion section.
Our participants were mostly young females, and we did not control for psychiatric diagnoses, therefore, we have no data on the diagnoses. We have now indicated this as a limitation and now consider this further when discussing the findings.
Methods
The study used the Perth Alexithymia Questionnaire-Short Form (PAQ-S) and a battery of self-report questionnaires assessing several key mental health outcomes (i.e., anxiety and depression symptoms, somatic complaints, well-being).
Results
The prevalence of alexithymia of the study population is very low when compared to studies of general populations. This probably shows that our sample represent mostly healthy young adults. The results showed that the relative and absolutely stability of alexithymia was high.
Reply: We agree that absolute and relative stability of alexithymia might be related to the fact that our sample represented mostly young healthy females. As we noted several differences in other mental health variables across time, with no differences in alexithymia, in our view, this can be also treated as an indicator of alexithymia stability in this non-clinical study. We have now added further details on this in the discussion section.
Discussion
The interpretations are justified by the results, and the relevant findings are discussed in relation to prior research. However, a more critical approach is needed, especially considering the sample size and the age and gender distribution. I would also appreciate a more critical examination of the concepts of absolute and relative stability. To what extent can these findings be generalized, and to which population.
Reply: In the discussion, we have now strove to present different interpretations of these data. We have further elaborated on the limitations, especially related to the characteristics of the study sample, and added details about the interpretations related to concepts of absolute and relative stability. We have also explained the extent of generalizability of these results.
Reviewer 2 Report
Comments and Suggestions for Authors
I do not have major comments, my only comment is to re-write the manuscript according to the instruction for authors.
Please, re-write the manuscript as the instruction for authors, although is not mandatory, it is better for the Reviewer evaluation of the manuscript. Also, it can mean that the authors are involving with changing the manuscript, because the current format of reference and citations is for a specific publisher.
The abstract I would recommend to include more crude data, and p values. The rest of the manuscript for me is unremarkable.
Author Response
We would like to thank the editor and the reviewers for their positive and encouraging feedback on our submission. The constructive comments of reviewers helped us to significantly improve the quality of our submission. We have been through all comments one by one, edited the manuscript in detail, and added new material where required. We hope the editor and reviewers find the revised version of the manuscript clear and suitable for publication in the Journal of Clinical Medicine. All changes made are highlighted in red (in the replies and the revised paper).
Reviewer 2
I do not have major comments, my only comment is to re-write the manuscript according to the instruction for authors.
Please, re-write the manuscript as the instruction for authors, although is not mandatory, it is better for the Reviewer evaluation of the manuscript. Also, it can mean that the authors are involving with changing the manuscript, because the current format of reference and citations is for a specific publisher.
The abstract I would recommend to include more crude data, and p values. The rest of the manuscript for me is unremarkable.
Reply: We have now edited our paper according to the instruction for authors. The reference format has been now edited. Also, as suggested, we have now added specific statistical numbers to the abstract.